# The Driving Factors of EMU Government Bond Yields: The Role of Debt, Liquidity and Fiscal Councils

## Anastasios Pappas [1] and Ioannis Kostakis [2,*]

[1]    Hellenic Fiscal Council, Greece and Department of Economics, National and Kapodistrian University of Athens, 15772 Athens, Greece; taspappas@econ.uoa.gr

[2]    Hellenic Fiscal Council, Greece and Department of Home Economics and Ecology, Harokopio University of Athens, 17676 Kallithea, Greece

[*]    Correspondence: ikostakis@hua.gr; Tel.: +30-210-95649216

**Abstract:** This study presents empirical evidence about the determinants of long-term government bond yields for 19 economies of the European Monetary Union (EMU) over the period 1995–2018 within a multivariate panel framework. The fixed effects estimators reveal that the relationship between public debt to the GDP ratio and yields is non-linear. We observe a threshold, which is determined to be at the area 90% of the ratio of public debt to GDP. Beyond that, area government borrowing costs increase as the public debt rises. Furthermore, we find evidence that a GDP decline and the downgrades of sovereign ratings increase the costs of government borrowing. In contrast, the operation of independent fiscal institutions helps to reduce government's debt risk premium. Finally, liquidity in the Euro area plays a significant role on yields determination. The results remain robust when the dynamic instrumental variable fixed effect (FE-2SLS) and dynamic panel least square dummy variable corrected (LSDVC) estimators are employed. Empirical findings suggest important policy implications for the ongoing Covid-19 crisis for the EMU.

**Keywords:** EMU; government bond yields; public debt; growth; liquidity; fiscal councils

**JEL Classification:** C23; E43; E62; G10

## 1. Introduction

The current pandemic crisis involving the "Covid-19" disease will challenge the strengths of the economies globally. There is a broad consensus among the forecasts that most economies all over the world will be hit by recessions. Especially, for the European Monetary Union (EMU) economies, the recession for 2020 is estimated by the European Central Bank to be within a range of 5%–15%. The significant slowdown of economic activity of the Euro-area countries will put further challenges on their fiscal positions and it is possible for upwards pressure governments' borrowing costs to develop. This scenario may put further fiscal constraints to economies, especially those with a high level of public debt. On the other hand, a substantial fiscal stimulus is considered to be necessary in order to mitigate the economic consequences of the crisis.

In this context, the debate about the driving factors of the government bond yields is very timely. If, indeed, the stock of public debt of the Euro-area countries is a significant factor of the determination of the cost of long-term public finance, we may expect—ceteris paribus—a considerable rise of the yields of long-term government bonds. This may further fuel the accumulation of public debt and further increase the ratio of public debt to GDP. The expected decrease of the euro-area economies' GDP may create an upward spiral of the public debt to GDP ratio.

A further consideration of the current paper is the turning point where the increase of public debt leads to an increase of the bond yields. Thus, we do not only investigate the nexus between public debt and long-term government bond yields in the Eurozone, but also, if a particular threshold exists where public debt affects the yields more.

Moreover, the role of macroeconomic performance, liquidity, sovereign credit rating and the electoral cycle are examined as possible determinants of long-term government bond yields of the euro-area economies. Last but not least, the link between the level of the yields and the operation of the European independent fiscal organizations (IFIS) is also investigated. This link has limited empirical investigation as far as the euro-area countries are concerned.

The rest of the paper is structured as follows: Section 2 includes a review of the relevant literature. In Section 3, the variables and the data set are discussed. Section 4 presents the econometric methodologies. Section 5 discusses the empirical results. Finally, Section 6 presents concluding remarks and policy implications.

## 2. Literature Review

The research on the determinants of long-term government bond yields focuses mainly on four categories of explanatory variables. The first category focuses on the fiscal situation within an economy and especially on the amount of public debt that the economy has accumulated as a percentage of the GDP. The ratio of public debt to GDP is considered as a measure of the ability of a government to service its debt. According to this view, with a larger ratio, the possibility of a sovereign default increases. In an early paper, Elmendorf and Mankiw (1999) reviewed literature on the macroeconomic effects of government debt. Among other concerns, they support the view that a high government debt could make an economy more vulnerable to a crisis of international confidence. In that case, a capital flight may cause a sharp rise of the cost of government financing. Furthermore, Codogno et al. (2003) assessed the importance of credit risk represented by the debt-to-GDP ratio for 10 EMU countries. They found that movements in yield differentials on the eurozone government bonds are explained by their debt-to-GDP ratios relative to Germany. Attinasi et al. (2009) found for 10 euro-area countries that higher government debt ratios relative to Germany contributed to higher government bond yield spreads. The view that fiscal indebtedness propels government bond yields is also presented in more recent papers (Ghosh et al. 2013; Afonso and Rault 2015; Paniagua et al. 2016).

In contrast however, other researchers do not share the view that long-term government yields movements are significantly related to fiscal fundamentals. Di Cesare et al. (2012) observe that the spreads vis-à-vis the German Bund have risen to levels that are significantly higher than what could be justified by the trends in the debt-to-GDP ratios for several European countries. De Grauwe and Ji (2012) find that yields of euro-area economies do not relate to changes in debt-to-GDP ratios. Georgoutsos and Migiakis (2013) found little evidence that fiscal variables are the main determinants of sovereign spreads for ten EMU countries. Furthermore, Moro (2014) while examining the recent European crisis, underlines the fact that a high level of public debt is not a problem per se, as long as the government is able to refinance itself and roll over its debt. Thus, the crisis was considered mainly as a competitiveness and growth crisis that has led to structural imbalances within the euro area.

Additionally, macroeconomic fundamentals may be linked to government bond yields volatility. Alexopoulou et al. (2010) suggest that macroeconomic fundamentals drive the long-term sovereign bond spreads of eight new EU countries. These countries' current account balances, exchange rates, inflation rates and the degree of trade openness play an important role in their ability to get access to long-term finance. Arghyrou and Kontonikas (2012) find that, especially after August 2007, a marked shift in market pricing behavior of the government bonds for EMU countries has been driven by macro-fundamentals.

However, other research suggests that, especially during crises episodes, macroeconomic factors are becoming less significant explanatory variables of government bond yields (Ebner 2009; De Grauwe and Ji 2012). Beirne and Fratzscher (2013) argue that the current account balance of the euro

area countries had little explanatory power for sovereign risk, especially before the global economic crisis of 2008. Seremetis and Pappas (2013) found no evidence that macroeconomic variables, such as the industrial production and consumer price index, affect the government bond yield spreads of six over-borrowed European countries.

Other literature considers liquidity factors as important determinants of government bond yields. Dailami et al. (2008) suggest that the US interest rate policy has a significant effect on the determination of credit spreads on emerging market bonds over US benchmark treasuries. Thus, the tightening of liquidity caused by an increase of US short-term interest rates may cause a steep increase of emerging market spreads. Manganelli and Wolswijk (2009) also found that spreads between euro area government bond yields are related to short-term interest rates, which are in turn related to market liquidity. More recently, Febi et al. (2018) investigated the relationship between liquidity risks and yield spreads for both green and conventional bonds. Their results reveal that increased liquidity is negatively related to the yield spread. Huchet and Serbini (2018), based on a panel of Member States of the Eurozone between 2008 and 2013, concluded that monetary policies decrease sovereign spreads by providing liquidity to interbank markets through collateral enlargements. The important role of liquidity for the de-escalation of government bond yield spreads in the Euro area is underlined in the paper of Afonso and Jalles (2019a). Their empirical evidence supports the argument that ECB's expansionary monetary intervention did contribute to containing sovereign yield spreads. Very recently the research of Afonso et al. (2020) confirmed that the announcements of the ECB's key interest rates together with the longer-term refinancing operations (LTROs) and liquidity injections contribute significantly to reduction of sovereign yield spreads of EMU countries.

Moreover, international risk aversion coupled with contagion effects may explain the overreaction of the market participants, which in turn escalates long-term government bond yields. Such an overreaction has been observed after the global economic crisis of 2007–2008 (De Grauwe and Ji 2012; Seremetis and Pappas 2013). Similar conclusions are drawn from the paper of Gruppe et al. (2017). They support the view that the introduction of the euro has eliminated exchange rate risk for financial transactions among those countries that have decided to join the EMU. Thus, before the recent European crisis, sovereign credit risk was not a crucial factor for the determination of the government bond yields. However, after the outbreak of the crisis, investors seem to fear sovereign credit risk and probably even redenomination risk that boosted the yields of the most vulnerable European economies; this result has also been found recently by Tholl et al. (2020). Furthermore, contagion effects may increase the volatility of government bond yields and push them to levels that cannot be justified by the economic fundamentals (Segoviano Basurto et al. 2010; Arghyrou and Kontonikas 2012; Silvapulle et al. 2016). During these adverse situations, the deterioration of sovereign credit ratings of the rating agencies may drive government bond yields even higher. Afonso et al. (2012) showed that EU government bond yield spreads were significantly affected by changes in the ratings of rating agencies (Standard & Poor's, Moody's, Fitch). Particularly in the case of negative announcements, this effect was stronger. A considerable amount of empirical research concurs with this view of a significant role of sovereign rating agencies for the determination of government bond yields (Reusens and Croux 2017; Huchet and Serbini 2018; Capelle-Blancard et al. 2019).

Finally, the recent enhanced European fiscal framework, with independent fiscal institutions monitoring the proper implementation of fiscal rules, may enhance fiscal discipline and reduce the governments' borrowing costs. There is some empirical evidence which affirms such a relation (Maltritz and Wüste 2015; Afonso and Jalles 2019b).

## 3. Variables and Data Set

We examined three different sets of variables; the fiscal variables focused exclusively on the public debt to GDP ratio (Debt). From a theoretical point of view, the public debt to GDP ratio could have a positive relation to borrowing costs, since the credit risk of an economy increases and the possibility for a default is higher (Gruber and Kamin 2012; Afonso and Jalles 2019a). Thus, a positive

sign is expected for describing the relation between the public debt to GDP ratio and government bond yields. We also included the public debt to GDP ratio variable in quadratic terms (Debt2) to investigate potential nonlinear effects of public debt to government bond yields (Ardagna et al. 2007; De Grauwe and Ji 2012; Filbien et al. 2013). By introducing non-linearity in the model, it is assumed that the economy's credibility becomes more vulnerable after a particular threshold of stock of public debt compared to the size of the economy. Beyond this "turning point", a capital flight may become more severe and boost government bond yields in higher levels.

The second set of variables aimed to control for the role of macroeconomic fundamentals in the determination of long-term government bond yields. In this respect, a strong real growth of the GDP may reduce the risk premium associated with high public debt, attract investors and induce capital inflows, resulting in a reduction of a government's borrowing costs (Eichengreen and Mody 2000; Reusens and Croux 2017). In contrast, high inflation and/or current account deficits may affect an economy during financial turbulence and may be associated with an increase of government bond yields (Alexopoulou et al. 2010; Buti and Carnot 2012). Thus, the growth of the real GDP (Growth), the rate of change of the harmonized index of consumer prices (HICP) and the current account balance (CA) as a percent of GDP were incorporated into the basic model as control variables.

The third set of variables controls for liquidity, sovereign ratings, electoral cycle, crisis effects and the role of the Independent Fiscal Institutions (IFIs). As proxies for liquidity conditions of the euro area, we used two variables: the main refinancing operations rate (MRO) of the European Central Bank and the growth of money supply calculated by the broad measure of M3 (M3). The expected sign of the ECB's main refinancing operations rate was positive since the yields are expected to move in the same direction with the interest rate, while the expected sign of the money supply was negative, since the monetary expansion may be associated with lower yields and vice versa (Ardagna et al. 2007).

In order to investigate the effects of sovereign credit ratings on the government bond yields, an index was constructed based on the sovereign rating scores that the Fitch rating agency applies (Fitch). We modeled end-of-year sovereign credit ratings, assuming that the frequency of the within-year rating change is small. The expected sign of the Fitch's variable coefficient is negative since the downgrade of the credibility of a country may result in a yields increase (Reusens and Croux 2017; Capelle-Blancard et al. 2019).

Furthermore, we constructed dummies to capture the effects of the electoral cycle, the Eurozone debt crisis and of the operation IFIs on the government bond yields dynamics. Electoral uncertainty may be captured by a dummy variable by placing a value of one at the year of national elections and zero otherwise (Elections). Theoretically, an election year may increase yields since investors may expect less fiscal discipline during that year (Vaaler et al. 2005; Li et al. 2013). A dummy variable with the value of one at 2012—the most turbulent year of the Eurozone debt crisis—and zero otherwise was constructed to capture the effects of international risk aversion (Ecrisis).

Finally, as a control variable, we tried to model the role of IFIs/Fiscal Councils by incorporating another dummy variable (Councils). The dummy takes the value of one by the year of the establishment of the institution and zero otherwise. Theoretically, the effect of the existence and the operation of fiscal councils may enhance not only the fiscal discipline but also the confidence of investors for government's debt securities, resulting in lower borrowing costs. Thus, a negative sign was expected for the IFI's dummy.

As the dependent variable, 10-year government bond yields (ltgb) were incorporated into the model. The variables were calculated on a yearly basis for an unbalanced panel of the 19 countries of Euro-area over the relevant period (1995–2018). The Euro-area was taken as an example for our investigation, as these countries have a single currency while its financial markets are highly interrelated and operate within the same fiscal framework. Furthermore, the EMU area had been hit by a major crisis, which strongly affected government bond yields. The source of this data was Eurostat. Table 1 presents the set groups of quantitatively used variables, while also providing information about its definition and the expected sign.

**Table 1.** Descriptive statistics, expected signs and definitions of the variables used.

| Variables | Mean | St. dev. | Min | Max | Expected Sign | Definition |
|---|---|---|---|---|---|---|
| Ltgb | 4.1 | 2.6 | −0.3 | 22.5 | | Long-Term Government Bond Yields: 10-year |
| Debt | 62.2 | 36.4 | 3.8 | 181.2 | + | Government consolidated gross debt (% of GDP) |
| Debt2 | 5194.2 | 5653.1 | 14.4 | 32,833.1 | + | Square term of government consolidated gross debt (% of GDP) |
| Growth | 2.7 | 3.7 | −14.8 | 25.2 | - | Growth rate of real GDP, Chain linked volumes, 2010 (%) |
| HICP | 2.5 | 2.6 | −1.7 | 24.7 | + | Harmonized inflation rate, (%) |
| CA | −0.7 | 5.8 | −20.7 | 12.6 | + | Total Current Account Balance for each country (% of GDP) |
| MRO | 3.0 | 1.8 | 0.3 | 5.9 | + | Main Refinancing Operations rate of the ECB |
| M3 | 1.6 | 0.6 | −0.01 | 2.4 | - | Money supply measured by M3 for the Euro Area (Growth rate previous period) |
| Fitch | 82.0 | 18.1 | 15 | 100 | - | Fitch credit ratings, Index |

## 4. The Econometric Models

In view of the above presented discussion, we estimated a country bonds yields equation using annual data for Eurozone economies over the period of 1995–2018. To investigate the behavior of bond yields growth, a fixed effects estimator was used which determines the significance of the public debt, macroeconomic indicators and other control variables on the bond-yields level. The fixed effects specification was as follows:

$$y_{i,t} = \beta_0 + \beta' X_{i,t} + \mu_\iota + e_{i,t} \tag{1}$$

where $y_{i,t}$ is the dependent variable in country $i$ at time period $t$; $\beta_0$ is the constant term; $X$ is the vector of independent variables that were used in the empirical analysis; $\beta'$ represents the estimated coefficients for each variable; $\mu_\iota$ is the unobserved heterogeneity and $e_{i,t}$ are the idiosyncratic terms. Regarding estimation tests, even if the majority of the literature (Arghyrou and Kontonikas 2012; De Grauwe and Ji 2012; Huchet and Serbini 2018; Capelle-Blancard et al. 2019) indicates that a fixed effects estimator is assumed to be conventionally more appropriate than a random effects for macroeconomic datasets, we performed the Hausman (1978) specification test in our analysis. The null hypothesis is that a random effects model is preferred (Greene 2008). Our result clearly rejected the null hypothesis, indicating that a fixed effects estimator is indeed more appropriate for the present empirical analysis. By doing so, the within-country variation was removed. In a vector form, the previous equation is expressed for countries $i$ as:

$$
\begin{bmatrix} y_{i,1} \\ y_{i,2} \\ . \\ . \\ y_{i,t} \end{bmatrix} = \begin{bmatrix} \mu_i \\ \mu_i \\ . \\ . \\ \mu_i \end{bmatrix} + \begin{bmatrix} x_{1i,1} & \cdots & x_{ki,1} \\ \vdots & \ddots & \vdots \\ x_{1i,t} & \cdots & x_{ki,t} \end{bmatrix} + \begin{bmatrix} \beta_1 \\ \beta_2 \\ . \\ . \\ \beta_k \end{bmatrix} + \begin{bmatrix} e_{i,1} \\ e_{i,2} \\ . \\ . \\ e_{i,t} \end{bmatrix}. \tag{2}
$$

Fixed effects estimation removes the effect of the time-invariant unobserved heterogeneity, while the slope coefficient remains the same. However, the intercepts are different because of differences in cross section-specific effects and because $\mu_i$ is allowed to be correlated with other regressors. Nevertheless, it is possible that the disturbance terms are different across those subsets (groupwise heteroscedasticity) that can cause biased standard error estimates. The modified Wald test was used leading to a strong rejection of the null hypothesis, which indicates the presence of

groupwise heteroscedasticity in our first fixed effects models. At the same time, serial correlation and cross-sectional dependency might also lead to biased standard errors estimates, giving an incorrect interpretation of results. To detect the presence of autocorrelation in the dataset, the Wooldridge (2002) test was run, the results of which implied that the model needs to account for autocorrelation. Lastly, for detecting cross-sectional dependence the Pesaran test (2004) was used, for which the results suggest that cross-sectional dependence is present in the data. Thus, to deal with these issues, the present study used Driscoll and Kraay (1998) standard errors in the fixed effect estimators that proved to be robust against heteroscedasticity, serial-correlation and cross-sectional dependence.

Furthermore, it is meaningful to ensure that the results are robust across different specifications. Therefore, since most of macroeconomic relations involve dynamic adjustments, we proceeded by employing dynamic specifications. However, we are aware that macroeconomic panel data models suffer from endogeneity, as the lagged dependent variable is correlated with error terms, leading to several econometric issues. To remedy this issue, two different approaches of dynamic panel estimators were employed, based on the fact that (a) endogeneity issues are present, (b) there is unobserved heterogeneity and (c) there is possible heteroskedasticity and autocorrelation within individual units' errors. Equation (3) presents the dynamic relationship among the rest of the independent variables:

$$y_{i,t} = \beta_0 + \sum_{j=1}^{p} \beta_j y_{i,t-j} + \beta X_{i,t} + e_{i,t} \tag{3}$$

$$e_{i,t} = \mu_i + u_{i,t} \tag{4}$$

where the error term $e_{i,t}$ is decomposed into the unobserved time-invariant heterogeneity $\mu_i$ and the idiosyncratic error component $u_{i,t}$. The one-way dynamic error component can be specified as follows:

$$y_{i,t} = \beta_0 + \beta_1 y_{i,t-1} + \beta' X_{i,t} + u_{i,t}. \tag{5}$$

Equation (5) incorporates both the long-run equilibrium relation and the short-run dynamics. However, it is known that the OLS estimator produces inconsistent results, as the lagged dependent variable creates autocorrelation since it is correlated to the unobserved heterogeneity component (Nickell 1981)[1]. In order to deal with this issue, 2SLS instrumental variable fixed effects estimator produces consistent estimates, particularly for macro-panel data. The recently proposed Least Square Dummy Variable Corrected (LSDVC) dynamic panel estimator using Anderson-Hsiao estimator and Monte-Carlo experiments (Kiviet 1995) are also employed. More specifically, to resolve the endogeneity bias issue, we followed the instrumental variable developed by Anderson and Hsiao (1982) within an estimator[2] that is consistent for the coefficients of the time-varying covariates. Furthermore, it should be noted that the consistency of the instrumental variable approach is mainly based on the validity of the instruments used in the estimated equation along with the absence of second-order serial correlations in the idiosyncratic term. Thus, tests for both under-identification (LM-statistic) and weak identification (Cragg–Donald) restrictions and for second-order serial correlation were employed. All tests supported the abovementioned estimated IV dynamic models, implying that the instruments used are valid and that the error term does not exhibit second-order serial correlation.

Next, we employed Bruno's (2005a) formula that is accurate as Kiviet's (1995) corrected LSDV estimator. Based on Judson and Owen (1999), it behaves better in terms of both root mean square errors and bias than the IV Anderson–Hsiao formula for a panel with a small number of cross-sectional units, while it also computes the bias correction for unbalanced dynamic panels, as is our case here.

---

[1]　$y_{i,t}$ is a function of unobserved effects, so $y_{i,t-1}$ is also a function of unobserved effects and it is correlated with the error term. A fixed effects estimator solves this issue by wiping out the unknown individual effects through within transformation of the data. In particular, as time becomes very large, the regressors become uncorrelated.

[2]　For instrumenting $y_{i,t-1}$ we used a 2-period lag value of the dependent variable $y_{i,t-2}$.

At the same time, the statistical significance of the estimated coefficients was tested using bootstrapped standard errors using 500 iterations (Bruno 2005b). Lastly, all the models presented the robust standard errors that have been calculated from the heteroskedastic-consistent variance-covariance matrix, as homoskedasticity was rejected.

## 5. Empirical Results

The empirical results for the panel of the Eurozone economies are presented in Table 2. The first three columns present the results based on our baseline methodology of the fixed effects estimator. We decomposed our analysis into three different models, namely (a) the benchmark fiscal model (Debt and Debt2), (b) the expanded model with macroeconomic variables (Growth, HICP, CA) and (c) the expanded model, which with the exception of fiscal and macroeconomic effects controls for liquidity conditions (MRO, M3), for European debt crisis effects (Ecrisis)[3], for electoral cycle effects (Elections), for sovereign credit ratings effects (Fitch) and for fiscal councils effects (Councils) on long-term government bond yields. The next three columns (4–6) present the empirical results of the same models but through using dynamic instrumental variable fixed effect estimator, while the last three columns (7–9) show the revealed results based on the recent-proposed (LSDVC) dynamic panel estimator using the Anderson–Hsiao estimator and Monte–Carlo experiments.

The non-linear specification seems to better explain the behavior of government bond yields compared to the accumulation of the public debt. Thus, following previous literature (Ardagna et al. 2007; De Grauwe and Ji 2012; Filbien et al. 2013), we added to our models the square of the debt variable, in order to investigate the existence of a non-linear relationship between public-debt to GDP ratio and government bond yield. Based on this theory, the rationale for including the square term of the debt-ratio variable is that bond yields may react differently after a certain threshold-debt-to-GDP level. The quadratic term of public debt to GDP (Debt2) remains highly significant for all the three specifications of the fixed-effect model. The positive sign of the variable implies a link between the rise in public debt and government borrowing cost increase. However, as other relative literature has shown, beyond a relative high turning point of the public debt to GDP ratio, the government borrowing cost increases. According to our calculations, the threshold for the Euro area economies is around the area of 90%. The negative sign of the debt variable supports the aforementioned result since the accumulation of public debt before the threshold seems to be associated with a lower cost of public financing. These results remain robust for all the different specifications and models.

As far as the macroeconomic variables are concerned, GDP growth has a significant negative relation with long-term government bond yields, while the inflation rate has a positive relation. These results are consistent with previous empirical research, which relates improved macroeconomic performance, in particular strong growth, with lower yields. In contrast, our results find no significant relation between the current account balance and long-term government bond yields. Our evidence shows that the current accounts deficits of the Euro area economies do not push yields higher and vice versa.

---

[3]    The *Ecrisis* dummy variable indicates whether financial crisis impacts are present in the given period affecting government bond yields. However, this approach has the limitation that, as Gruppe et al. (2017) have already indicated, the impact of a European shock can be also endogenously determined.

**Table 2.** Empirical results based on fixed effects (FE), fixed effects two-stage least squares (FE 2sls) and Least Square Dummy Variable Corrected (LSDVC) estimators.

| Variables | (Model 1) FE | (Model 2) FE | (Model 3) FE | (Model 4) FE 2sls | (Model 5) FE 2sls | (Model 6) FE 2sls | (Model 7) LSDVC | (Model 8) LSDVC | (Model 9) LSDVC |
|---|---|---|---|---|---|---|---|---|---|
| L.ltgb | | | | 0.644 *** | 0.665 *** | 0.040 | 0.803 *** | 0.768 *** | 0.483 *** |
| | | | | (0.063) | (0.0407) | (0.100) | (0.039) | (0.035) | (0.041) |
| Debt | −0.108 *** | −0.0936 *** | −0.0595 *** | −0.0475 *** | −0.0309 ** | −0.065 *** | −0.0359 ** | −0.0302 ** | −0.040 *** |
| | (0.028) | (0.017) | (0.017) | (0.017) | (0.0164) | (0.014) | (0.015) | (0.015) | (0.014) |
| Debt2 | 0.000555 *** | 0.000516 *** | 0.0002539 *** | 0.000205 *** | 0.000150 | 0.0002398 *** | 0.000183 ** | 0.000179 ** | 0.0001353 ** |
| | $(9.36 \times 10^{-5})$ | $(8.42 \times 10^{-5})$ | $(7.00 \times 10^{-5})$ | $(2.10 \times 10^{-5})$ | $(1.01 \times 10^{-5})$ | $(6.86 \times 10^{-5})$ | $(8.18 \times 10^{-5})$ | $(7.50 \times 10^{-5})$ | $(6.85 \times 10^{-5})$ |
| Growth | | −0.192 *** | −0.107 *** | | −0.129 *** | −0.111 *** | | −0.119 *** | −0.111 *** |
| | | (0.061) | (0.033) | | (0.031) | (0.018) | | (0.025) | (0.023) |
| HICP | | 0.367 *** | 0.110* | | 0.257 *** | 0.066 | | 0.252 *** | 0.177 *** |
| | | (0.109) | (0.053) | | (0.048) | (0.049) | | (0.058) | (0.053) |
| CA | | −0.00326 | 0.017 | | −0.0185 | 0.005 | | −0.0206 | −0.005 |
| | | (0.058) | (0.027) | | (0.019) | (0.019) | | (0.025) | (0.022) |
| MRO | | | 1.034 *** | | | 0.975 *** | | | 0.490 *** |
| | | | (0.175) | | | (0.144) | | | (0.098) |
| M3 | | | −0.280 *** | | | −0.261 *** | | | −0.136 *** |
| | | | (0.045) | | | (0.042) | | | (0.038) |
| Ecrisis | | | 0.459 ** | | | 0.513 * | | | 0.054 |
| | | | (0.227) | | | (0.313) | | | (0.337) |
| Elections | | | 0.258 | | | 0.204 | | | 0.217 |
| | | | (0.169) | | | (0.144) | | | (0.171) |
| Fitch | | | −0.103 *** | | | −0.109 *** | | | −0.064 *** |
| | | | (0.021) | | | (0.014) | | | (0.013) |
| Councils | | | −0.801* | | | −0.712 *** | | | −0.338 |
| | | | (0.434) | | | (0.268) | | | (0.315) |
| Constant | 8.094 *** | 6.882 *** | 13.988 *** | | | | | | |
| | (1.348) | (0.744) | (2.500) | | | | | | |
| | | | | | | | | | |
| Threshold | 97.5% | 90.8% | 118.1% | 115.9% | 103.0% | 135.2% | 98.1% | 84.4% | 146.4% |
| Observations | 408 | 390 | 388 | 372 | 370 | 370 | 390 | 384 | 364 |
| R-squared | 0.081 | 0.243 | 0.739 | 0.617 | 0.699 | 0.728 | | | |
| Number of countries | 19 | 19 | 19 | 19 | 19 | 19 | 19 | 19 | 19 |
| Hausman | 29.34 *** | 34.8 *** | 46.67 *** | | | | | | |
| LM statistic | | | | 196.2 *** | 208.6 *** | 84.7 *** | | | |
| Cragg-Donald | | | | 437.8 *** | 505.3 *** | 107.7 *** | | | |

Notes: Driscoll and Kraay (1998) standard errors in parentheses in the first three models and robust standard errors for the rest of the models; *** $p < 0.01$, ** $p < 0.05$, * $p < 0.1$.

Furthermore, liquidity within the Euro area is associated significantly with government borrowing costs. The more liquidity is directed into the Euro area, the lower the long-term government bond yields will be. Both proxies of liquidity conditions (MRO and M3) were found to explain government bond yields movements of the Euro area economies for the last 20 years. More specifically, the main refinancing operations rate (MRO) of the European Central Bank is positively associated with the long-term government bond yields, since a reduction of the facility's interest rate increases liquidity and pushes government bond yields to lower levels. The growth of money supply (M3) is also linked to yields movements. According to our results, the increase of the money supply drives yields lower and the opposite also occurs; the contraction of liquidity induces higher yields.

The sovereign credit ratings are also found to be significant determinants of the long-term government bond yields of the Euro area. The index of Fitch ratings that we constructed is linked with the fluctuations of the yields. The downgrades of sovereign creditability increases the cost of government financing and the opposite trend is also valid. The European debt crisis was also found to be associated with increased yields. Furthermore, we found no significant relation between the electoral cycle (Elections) and the government bond yields in the Euro area.

Interestingly, the dummy variable used as a proxy for the role of IFIs/Fiscal Councils was found to be a significant determinant of the government bond yields fluctuations. The negative sign of the variable (Councils) implies that the establishment and the operation of such institutions is associated with lower government borrowing costs. The rationale that explains this result may be that a strong fiscal framework, with fiscal councils acting as fiscal rules' "watchdogs", reinforces investors' confidence and increases the reduction of risk premiums on long-term government bonds.

## 6. Conclusions

The findings of this paper provide important policy implications for the Euro area economies and in particular, during the evolving economic crisis of Covid-19, as far as the determinants of long-term government bond yields are concerned. We found that the effect of public debt to GDP ratio on long-term government bond yields is non-linear, thus the response of the yields is only positive when the ratio is above the given threshold of roughly 90%. Therefore, the Eurozone countries which have public debt to GDP ratios within or above the area of 90% may encounter a—ceteris paribus—significant increase of their government borrowing cost reflected as higher long-term government bond yields. Currently the public debt-to-GDP ratio of seven out of the 19 EMU countries exceeds 95%. These countries will face a considerable upward pressure to their public debt to GDP ratio that stems from the intense fiscal stimulus in order to avoid a deep recession.

According to our results, two additional factors may currently deteriorate the cost of government financing of the Eurozone economies: the (expected) deep decline of the GDP and the possible sovereign rating downgrades from the rating agencies. We found evidence that both significantly affect the fluctuations of long-term government bond yields of the Euro area countries. Thus, these two factors may put further stress on the cost of government financing and negatively impact the public debt to GDP ratio, widening the gap between Europe's North and South and putting the stabilization of the EMU under severe risk.

However, we found evidence that a significant liquidity increase within the Euro area will mitigate the aforementioned adverse effects on public debt cost and will prohibit further debt accumulation. In this respect, the response of the ECB to provide liquidity to the Euro system has helped the stabilization of the Eurozone government bond yields and enhanced the financial stability in Europe. These actions must be continued and become more intensified whenever necessary.

After the crisis, a path towards less public debt accumulation may be necessary to act as a financial shock absorber during "bad times". In this respect, the fiscal councils of the European countries must contribute to the improvement of fiscal discipline during the "good times".

**Author Contributions:** Conceptualization, A.P. and I.K.; methodology, A.P. and I.K.; software, I.K.; validation, A.P.; formal analysis, A.P. and I.K.; investigation, A.P. and I.K.; resources, A.P.; data curation, A.P. and I.K.; writing—original draft preparation, A.P. and I.K.; writing—review and editing, A.P. and I.K.; visualization, A.P. and I.K.; supervision, A.P.; project administration, A.P.; funding acquisition, None. All authors have read and agreed to the published version of the manuscript.

**Funding:** This research received no external funding.

**Conflicts of Interest:** The authors declare no conflict of interest.

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
