# Peer review of "The Driving Factors of EMU Government Bond Yields: The Role of Debt, Liquidity and Fiscal Councils"

_ijfs, doi:10.3390/ijfs8030053_

Round 1
Reviewer 1 Report
The research provides the comprehensive empirical study about the determinants of long-term government 12 bond yields for 19 economies of the European Monetary Union (EMU) over the period 1995–2018 within a multivariate panel framework. The authors show that the GDP decline 17 and the downgrades of sovereign ratings increase the costs of government borrowing. The deep decline of the GDP and the sovereign rating downgrades both affect significantly the fluctuations of long-term government bond yields of the Euro area countries. They suggest important policy implications for the ongoing Covid-19 crisis for the EMU. The paper’s main findings are interesting. However, there are still some questions or comments on which the authors should address.
(1) Why study European Monetary Union (EMU) government bonds? What are so special about them?
(2) In page 7, the authors state that the non-linear specification seems to better explain the behaviour of government bond yields compared to the accumulation public debt. The results in Table 2 are not enough to support the funding. The author is necessary to provide more powerful results to the reader.
Author Response
Reviewers’ comments and revisions
Please find attached our revised paper entitled “The driving factors of EMU government bond yields: The role of debt, liquidity and fiscal councils”. Following the reviewers’ comments, in this new revised version, we have incorporated all the referees’ suggestions. Please find below a detailed (point by point) list of changes.
Reviewer 1:
The research provides the comprehensive empirical study about the determinants of long-term government 12 bond yields for 19 economies of the European Monetary Union (EMU) over the period 1995–2018 within a multivariate panel framework. The authors show that the GDP decline 17 and the downgrades of sovereign ratings increase the costs of government borrowing. The deep decline of the GDP and the sovereign rating downgrades both affect significantly the fluctuations of long-term government bond yields of the Euro area countries. They suggest important policy implications for the ongoing Covid-19 crisis for the EMU. The paper’s main findings are interesting. However, there are still some questions or comments on which the authors should address.
*1. Why study European Monetary Union (EMU) government bonds? What are so special about them?
As proposed, we revised our manuscript giving also emphasis on the reasons that we chose to study EMU government bonds. The main reason as we also have incorporated in our revised manuscript, is that, based also on previous literature, we took as an example of our investigation a group of countries with same or similar characteristics (currency and highly interrelated financial markets that follow the same fiscal rules).
*2. In page 7, the authors state that the non-linear specification seems to better explain the behaviour of government bond yields compared to the accumulation public debt. The results in Table 2 are not enough to support the funding. The author is necessary to provide more powerful results to the reader.
Thank you for your valuable comment. Following the reviewer’s proposal, we revised the following paragraph giving more information about the reasons we followed non-linear specification for explaining the behavior of government bond yields. More specifically, we followed the De Grauwe and Ji (2012) methodological approach by adding to our models the square of the debt variable, in order to investigate the existence of a non-linear relationship between public-debt to GDP ratio and government bond yield.
Reviewer 2 Report
This paper is very interesting. I like the topic. However, I would suggest to make some changes. The the study could be published.
Most importantly, the literature review has to be imporved. The sovereign debt criris (Grecce…) should be discussed in more detail. I would suggest to cite the excellent review paper by Moro (2014). Moreover, given the importance of possible problems with structural change (see below) I would also cite the survey by Gruppe et al. (2017) – and possibly also some of the literature discussed there! With regard to the over- and underpricing of sovereign risk I would also cite the recent paper by Afonso, Jalles and Kazemi (2020) and some additional literature cited there. Moreover, with regard to Covid-19 and Italy the paper by Tholl et al. (2020) obviously should be cited.
Structural change is of some importance here. With regard to the methodology used in this paper I am not quite sure that a simple dummy variable can capture the effects of the EMU government debt crisis. Therefore, I would at least discuss the limitations of this approach. Probably the Gruppe et al. (2017) paper (and some literature cited there) can help!
Additionally, I have some problems with one sentence on page 7. I am not quite sure that economic theory does clearly predict a negative relationship between real GDP growth and interest rates. This is true for fixed income securities that are subject to credit risk. But with regard to the “risk-free” rate this is not necessarily true (lower demand for safe assets and more demand for capital in booms)!
New literature:
Afonso, A., Jalles, J. T., & Kazemi, M. (2020). The effects of macroeconomic, fiscal and monetary policy announcements on sovereign bond spreads. International Review of Law and Economics, (online first)
Gruppe, M. et al. (2017). Interest rate convergence, sovereign credit risk and the European debt crisis: a survey. The Journal of Risk Finance, 18, 432-442.
Moro, B. (2014). Lessons from the European economic and financial great crisis: A survey. European Journal of Political Economy, 34, S9-S24.
Tholl, J. et al. (2020). Bank funding and the recent political development in Italy: What about redenomination risk?. International Review of Law and Economics, (online first)
Author Response
Reviewers’ comments and revisions
Please find attached our revised paper entitled “The driving factors of EMU government bond yields: The role of debt, liquidity and fiscal councils”. Following the reviewers’ comments, in this new revised version, we have incorporated all the referees’ suggestions. Please find below a detailed (point by point) list of changes.
Reviewer 2:
This paper is very interesting. I like the topic. However, I would suggest to make some changes. Then the study could be published.
*3. Most importantly, the literature review has to be improved. The sovereign debt criris (Greece…) should be discussed in more detail. I would suggest to cite the excellent review paper by Moro (2014). Moreover, given the importance of possible problems with structural change (see below) I would also cite the survey by Gruppe et al. (2017) – and possibly also some of the literature discussed there! With regard to the over- and underpricing of sovereign risk I would also cite the recent paper by Afonso, Jalles and Kazemi (2020) and some additional literature cited there. Moreover, with regard to Covid-19 and Italy the paper by Tholl et al. (2020) obviously should be cited.
New literature:
Afonso, A., Jalles, J. T., & Kazemi, M. (2020). The effects of macroeconomic, fiscal and monetary policy announcements on sovereign bond spreads. International Review of Law and Economics, (online first)
Gruppe, M. et al. (2017). Interest rate convergence, sovereign credit risk and the European debt crisis: a survey. The Journal of Risk Finance, 18, 432-442.
Moro, B. (2014). Lessons from the European economic and financial great crisis: A survey. European Journal of Political Economy, 34, S9-S24.
Tholl, J. et al. (2020). Bank funding and the recent political development in Italy: What about redenomination risk?. International Review of Law and Economics, (online first)
Thank you very much for this comment. Following the valuable comment of the reviewer we revised our literature review analysis adding the proposed published papers.
*4. Structural change is of some importance here. With regard to the methodology used in this paper I am not quite sure that a simple dummy variable can capture the effects of the EMU government debt crisis. Therefore, I would at least discuss the limitations of this approach. Probably the Gruppe et al. (2017) paper (and some literature cited there) can help!
Thank you very much for this comment. Based on the valuable proposal of the reviewer, we revised the specific paragraph highlighting the reasons we used dummy variables to capture specific effects of the EMU government debt crisis. More specifically, based on previous research we took into account the impact of the financial crisis that has been incorporated into the model by using a financial crisis dummy. Methodologically, dummy variables can be added as independent variable using the time horizon of the available data. Alternatively, the dummies indicate whether there were financial crisis effects present in the particular period. However, based on the valuable comment of the reviewer, we added limitations for giving a clearer picture of the methodological approach that we followed.
*5. Additionally, I have some problems with one sentence on page 7. I am not quite sure that economic theory does clearly predict a negative relationship between real GDP growth and interest rates. This is true for fixed income securities that are subject to credit risk. But with regard to the “risk-free” rate this is not necessarily true (lower demand for safe assets and more demand for capital in booms)!
Thank you for your valuable comment. For a better clarification, we should indicate that we focus particularly to the economic theory for the relation between interest rates and growth and not to the effect of the interest rate fluctuations on to the financial assets.